# Application of PLGA-PEG-PLGA Nanoparticles to Percutaneous Immunotherapy for Food Allergy

**DOI:** 10.3390/molecules29174123

**Published:** 2024-08-30

**Authors:** Ryuse Sakurai, Hanae Iwata, Masaki Gotoh, Hiroyuki Ogino, Issei Takeuchi, Kimiko Makino, Fumio Itoh, Akiyoshi Saitoh

**Affiliations:** 1Faculty of Pharmaceutical Sciences, Tokyo University of Science, 2641, Yamazaki, Noda 278−8510, Chiba, Japan; sakurairyuse@gmail.com (R.S.); itakeuchi@jiu.ac.jp (I.T.);; 2Modality Research Group, BioPharma Research Institute, Kaneka Corporation Inc., 1−8 Miyamae-cho, Takasago-cho, Takasago-shi 676−8688, Hyogo, Japan; 3Faculty of Pharmaceutical Sciences, Josai International University, 1 Gumyo, Togane 283−8555, Chiba, Japan; 4Department of Gastroenterology, St. Marianna University School of Medicine, 2−16−1 Sugao, Miyamae-ku 216−8511, Kawasaki, Japan

**Keywords:** PLGA, PLGA-PEG-PLGA, nanoparticles, epicutaneous immunotherapy (EPIT), iontophoresis, allergen immunotherapy (AIT), food allergy, percutaneous absorption

## Abstract

Compared with oral or injection administration, percutaneous immunotherapy presents a promising treatment modality for food allergies, providing low invasiveness and safety. This study investigated the efficacy of percutaneous immunotherapy using hen egg lysozyme (HEL)-loaded PLGA-PEG-PLGA nanoparticles (NPs), as an antigen model protein derived from egg white, compared with that of HEL-loaded chitosan hydroxypropyltrimonium chloride (CS)-modified PLGA NPs used in previous research. The intradermal retention of HEL in excised mouse skin was measured using Franz cells, which revealed a 2.1-fold higher retention with PLGA-PEG-PLGA NPs than that with CS-modified PLGA NPs. Observation of skin penetration pathways using fluorescein-4-isothiocyanate (FITC)-labeled HEL demonstrated successful delivery of HEL deep into the hair follicles with PLGA-PEG-PLGA NPs. These findings suggest that after NPs delivery into the skin, PEG prevents protein adhesion and NPs aggregation, facilitating stable delivery deep into the skin. Subsequently, in vivo percutaneous administration experiments in mice, with concurrent iontophoresis, demonstrated a significant increase in serum IgG1 antibody production with PLGA-PEG-PLGA NPs compared with that with CS-PLGA NPs after eight weeks of administration. Furthermore, serum IgE production in each NP administration group significantly decreased compared with that by subcutaneous administration of HEL solution. These results suggest that the combination of PLGA-PEG-PLGA NPs and iontophoresis is an effective percutaneous immunotherapy for food allergies.

## 1. Introduction

The prevalence of food allergies has increased worldwide over the past few decades, with >10% infants reported to be affected in some countries [1]. The majority of food allergies are IgE-dependent reactions, wherein the pathogenic mechanism involves the interaction of the allergen-specific IgE receptor (FcεRI) complex expressed on mast cells and basophils with the reintroduced allergen, which releases inflammatory mediators and causes anaphylactic FcεRI. Non-IgE-mediated anaphylaxis also exists, whose mechanism may involve the activation of anaphylatoxin and IgG pathways in the complement system [2]. Anaphylaxis can cause various symptoms in the gastrointestinal tract, skin, and respiratory and cardiovascular systems, and in severe cases, it can lead to acute respiratory or cardiac arrest [3]. The basis of treatment is the elimination of the causative food and symptomatic treatment of the triggering symptoms; nonetheless, this is not a fundamental cure for allergy, and patients are constantly forced to eliminate foods or risk inducing anaphylaxis by accidental ingestion, which places a tremendous burden on them [4,5]. Allergen immunotherapy (AIT) triggers immune tolerance to allergens through regular and repeated exposure to disease-causing allergens, inducing the remission of allergic symptoms and preventing the onset of allergy [6]. The major AIT currently available is oral immunotherapy (OIT), which is highly effective but problematic because of the high incidence of adverse reactions and the rare possibility of severe symptoms, including unanticipated anaphylactic shock [7]. Sublingual immunotherapy (SLIT), involving the administration of allergenic foods via the sublingual route, is an existing method; however, data on its efficacy and safety remain limited [8].

In recent years, epicutaneous immunotherapy (EPIT) has received attention as a new AIT. EPIT has demonstrated efficacy in the treatment of allergies to cow’s milk and peanuts [8,9]. In AIT, ideal targets for application include sites where a large number of strong antigen-presenting cells have the potential for improved efficacy, and where allergen delivery can be minimized without the development of a vascular system. In EPIT, allergen delivery occurs through the “skin” that fulfills this requirement [10]. In the skin, keratinocytes, which are involved in immune responses, account for 90% of the epidermis, and antigen-presenting cells include Langerhans cells in the epidermal spinous layer and dendritic cells in the dermis. Langerhans cells and dendritic cells trigger immune responses to external stimuli and foreign substances entering the skin, enabling an effective allergen delivery through the skin [11]. The percutaneous route is also noninvasive and, unlike oral administration, not subject to pH or enzyme interference, allows for controlled drug release, and reduces patient burden, rendering it a simple and safe route of administration for children and elderly individuals [12]. Nevertheless, in the percutaneous absorption route, the physical defense of the stratum corneum in the skin is a barrier. In particular, substances with molecular weights exceeding 500 Da have low permeability through the stratum corneum [13,14], making percutaneous absorption difficult by simply applying the allergen, a high-molecular-weight protein, to the skin surface. Therefore, technologies have been developed in recent years for the delivery of allergens into the skin. Recent studies on EPIT have included freeze-dried allergens in patches [15], microneedling, which improves the percutaneous delivery of allergens by puncturing the stratum corneum with tiny biodegradable needles [16], lipid-based colloids containing allergens [17], laser microporation [18,19], and other approaches.

This study focused on a percutaneous absorption system that combines nanoparticulation of allergens with iontophoresis (IP), which actively promotes the percutaneous absorption of charged drugs by electric repulsive force. The advantages of nanoparticulation include the ability to control the release of drugs (antigens in this study), improve the physicochemical stability of drugs, avoid aggregation by drug solubilization, and target specific tissues and cells [20]. Moreover, nanoparticles (NPs) <200 nm are comparable in size to antigens, such as viruses, and thus increase drug uptake by antigen-presenting cells [21]. There are also reports of improved percutaneous absorption of macromolecular compounds, such as peptides and proteins, using IP [22], shorter percutaneous delivery times, and control of blood concentration by energizing conditions. In our previous study, hen egg lysozyme (HEL), a model antigen found in chicken egg white, was nanoparticulated using polylactic acid glycolic acid copolymer (PLGA)—a biodegradable, biocompatible polymer known for its safety and commercialization as a pharmaceutical product—and used in combination with IP [23]. The results demonstrated improved percutaneous absorption of HEL in ex vivo tests and a significant decrease in IgE antibody titer in in vivo tests [24]. Another study reported that NPs prepared using PLGA-block-poly (ethylene glycol)-block-PLGA triblock copolymer (PLGA-PEG-PLGA) had improved intracutaneous drug accumulation compared with that of PLGA NPs [25]. Moreover, research utilizing PLGA-PEG-PLGA have been seen an increase in recent years, including studies on AIT [26,27,28]. Therefore, the application of PLGA-PEG-PLGA nanoparticles in transdermal immunotherapy offers a promising approach to enhance therapeutic outcomes by increasing intracutaneous drug retention, highlighting substantial potential in the development of innovative treatments for food allergies. The aim of this study was to investigate the utility of PLGA-PEG-PLGA NPs, a new drug carrier for the transdermal immunotherapy of HEL. In this study, FITC-labeled HEL (FITC-HEL) was used to visualize the transdermal absorption pathway of the drug following administration and to observe its penetration within the skin. Furthermore, the delivery potential of HEL intradermally and the in vivo antibody titers in therapeutic experiments were investigated, and the results were compared between CS-modified PLGA NPs developed in a previous study [24] and PLGA-PEG-PLGA NPs.

## 2. Results

### 2.1. Evaluation of Physical Properties of Prepared Nanoparticles

The average particle size of the NPs, as measured by dynamic light scattering, was 100 ± 47 nm for HEL-loaded CS-PLGA NPs, 99 ± 49 nm for FITC-HEL-loaded CS-PLGA NPs, 105 ± 54 nm for HEL-loaded PLGA-PEG-PLGA NPs, and 87 ± 43 nm for FITC-HEL-loaded PLGA-PEG-PLGA NPs. Figure 1 shows the results of the volume size distribution of the NPs. No difference was observed in the average particle size between HEL and FITC-HEL after preparation. The physical properties of NPs are shown in Table 1. The zeta potential of the NPs in 5 mM NaCl solution was similar to that observed during drug administration, viz., 38.4 ± 1.9 mV for HEL-loaded CS-PLGA NPs and −7.2 ± 0.1 mV for HEL-loaded PLGA-PEG-PLGA NPs. The HEL content in the NPs was 8.0 ± 0.8% in the HEL-loaded CS-PLGA NPs group and 3.4 ± 0.4% in the HEL-loaded PLGA-PEG-PLGA NPs group. The HEL content in the PLGA-PEG-PLGA NPs was less than half that in the CS-PLGA NPs. The impact of the presence or absence of FITC labeling of HEL on the drug content could not be confirmed. The results of the HEL release test are depicted in Figure 2. The cumulative amount of HEL released from HEL-loaded PLGA-PEG-PLGA NPs was significantly higher at 0.5 and 1 h after the start of the study, indicating faster drug release from PLGA-PEG-PLGA nanoparticles compared to PLGA nanoparticles.

### 2.2. Ex Vivo Measurement of Intracutaneous HEL Accumulation 

Figure 3 shows the amount of intradermally stored HEL 2 h after the start of the study. The intracutaneous HEL accumulation in the FITC-HEL-loaded PLGA-PEG-PLGA NPs group (225.3 ± 72.6 μg/g) was more than two times higher than that in the FITC-HEL-loaded CS-PLGA NPs group (79.1 ± 20.0 μg/g).

### 2.3. Ex Vivo Observation of the Intradermal Permeation Pathway of FITC-HEL

Figure 4 illustrates the intracutaneous pathway of FITC-HEL in the dorsal skin sections of mice after the ex vivo test. Compared with the FITC-HEL-loaded CS-PLGA NPs group, the FITC-HEL-loaded PLGA-PEG-PLGA NPs group exhibited significantly higher delivery of FITC-HEL to the deep skin through the system adnexal pathway via hair follicles. This result correlates with the higher HEL storage observed in the FITC-HEL-loaded PLGA-PEG-PLGA NPs group, as described in Section 2.2 regarding the skin HEL storage test results.

### 2.4. Results of the Measurement of Antibody Titer in Blood after in Vivo Percutaneous Immunization Experiment

The anti-HEL IgG1 antibody titer of each treatment group at eight weeks after the start of treatment is depicted in Figure 5a, showing a significant improvement in the PLGA-PEG-PLGA NPs group compared with that in the positive control solution-SC group and CS-PLGA NPs group. Figure 5b shows the results of anti-HEL IgG2a antibody titers, indicating no significant difference between the CS-PLGA NPs group and PLGA-PEG-PLGA NPs group. However, in the PLGA NPs group, a significant increase was only observed in the naïve group, whereas in the PLGA-PEG-PLGA NPs group, a significant improvement was observed between the naïve group and the solution-SC group. Finally, Figure 5c shows the results of serum total IgE antibody titers, which were significantly reduced in each NP group compared with those in the solution-SC group. In summary, Although IgE antibody production was suppressed in each NP group, the IgG1 antibody titer significantly increased in the PLGA-PEG-PLGA NPs group compared with that in the PLGA NPs group.

## 3. Materials and Methods

### 3.1. Materials

Polylactic acid glycolic acid copolymer (PLGA; molecular weight: 10,000, monomer composition of dl-lactic acid/glycolic acid = 75/25) was purchased from Taki Chemical Co., Ltd. (Kakogawa, Hyogo, Japan). PLGA-block-poly (ethylene glycol)-block-PLGA triblock copolymer (PLGA-PEG-PLGA, molecular weight: 3500/4000/3500, monomer composition of dl-lactic acid/glycolic acid/ethylene oxide = 39/14/47) was kindly donated by Taki Chemical Co., Ltd. The monomer composition of PLGA and PLGA-PEG-PLGA was determined by dissolving the polymers in deuterated acetone and analyzing the results obtained from 1H NMR spectroscopy. The molecular weight was determined by dissolving the polymer in tetrahydrofuran, measuring it with gel permeation chromatography (GPC), and calculating the weight-average using a calibration curve generated with standard polystyrene. Lysozyme, from egg white (HEL, C_616_H_963_N_193_O_182_S_10_∙HCl), l-(+)- arginine (purity ≥ 98.0%), l-glutamic acid (purity ≥ 99.0%), dimethyl sulfoxide (DMSO, C_2_H_6_OS, purity ≥ 99.0%), trifluoroacetic acid (TFA, CF_3_COOH, purity ≥ 99.8%), paraformaldehyde (purity ≥ 94.0%), sodium chloride, potassium chloride, ethyl acetate, fluorescein-4-isothiocyanate (FITC, purity > 95%), urea (purity ≥ 99%), sodium hydrogen carbonate, sodium carbonate, 2-amino-2-hydroxymethyl-1,3-propanediol hydrochloride, and bovine serum albumin (BSA, fatty acid/IgG/protease-free) were purchased from Fujifilm Wako Pure Chemical Corp. (Osaka, Japan). Chitosan hydroxypropyltrimonium chloride, with a molecular weight of 200,000 (CS), was kindly donated by Katakura & Co-op Agri Corp. (Tokyo, Japan). Acetonitrile (CH_3_CN; Japanese Pharmacopoeia, United States Pharmacopoeia/National Formulary, European Pharmacopoeia) was purchased from Kanto Chemical Co., Inc. (Tokyo, Japan). Isoflurane for animals was purchased from Mylan Inc. (Pittsburgh, PA, USA). Atipamezole hydrochloride was purchased from Kyoritsu Seiyaku Corp. (Tokyo, Japan). Vetorphale^®^ 2.5 mL (butorphanol tartrate 5.0 mg/ mL) was purchased from Meiji Seika Pharma Co., Ltd. (Tokyo, Japan). Dormicam Injection^®^ 2 mL (midazolam 5.0 mg/ mL) was purchased from Maruishi Pharmaceutical Co., Ltd. (Osaka, Japan). Domitor^®^ 0.75 mL (medetomidine hydrochloride 1.0 mg/mL) was purchased from Nippon Zenyaku Kogyo Co., Ltd. (Fukushima, Japan). Anti-IgG1 (mouse, goat-ply, AP) and anti-IgG2a (mouse, goat-ply, AP) were purchased from Novus Biologicals (Centennial, CO, USA). The alkaline phosphatase yellow (p-NPP) liquid substrate system for ELISA ready-to-use solution was purchased from Sigma-Aldrich (St. Louis, MO, USA). Dulbecco’s phosphate-buffered saline (PBS) was purchased from Nissui Pharmaceutical Co., Ltd. (Tokyo, Japan). OTSUKA NORMAL SALINE was purchased from Otsuka Pharmaceutical Factory, Inc. (Tokushima, Japan).

### 3.2. Animals

Female BALB/c mice (six weeks old, purchased from Sankyo Labo Service Corporation, Inc., Tokyo, Japan) were used for all experiments. The animals had free access to food and water in an animal room maintained at a stable temperature of 23 °C ± 1 °C and relative humidity of 55% ± 5% under a 12-hour light/dark cycle (lights on at 8:00 a.m. and off at 8:00 p.m.). All the animal experiments were conducted in accordance with protocols approved by the Institutional Animal Care and Use Committee of Tokyo University of Science (approval no. Y21052, approval date 12 April 2022).

### 3.3. Preparation of HEL-Loaded PLGA and PLGA-PEG-PLGA Nanoparticle Formulations

NPs were prepared using the poor solvent diffusion method and selective solvation as described in previous studies [24,29,30]. NPs were prepared by adding 6 mg of HEL to 44 mg of PLGA or PLGA-PEG-PLGA, dissolving in 2 mL of the good solvent DMSO, and then pipetting the good solvent into 10 mL of the poor solvent, 1.0% (weight/volume (*w*/*v*)) arginine solution. For dialysis washing, 1 L of 0.01% (*w*/*v*) arginine solution was used as the outer solution of the dialysis membrane, which was constantly stirred at 400 rpm with a stirrer and replaced with a new outer solution after 1, 3, and 5 h for a total of 24 h. After dialysis washing, the NPs suspension in the tubes was collected and designated as HEL-loaded PLGA NPs and HEL-loaded PLGA-PEG-PLGA NPs. Moreover, HEL-loaded PLGA NPs were CS-modified to improve skin penetration by making them positively charged [31,32]. Specifically, HEL-loaded CS-modified PLGA NPs (HEL-loaded CS-PLGA NPs) were obtained by mixing the HEL-loaded PLGA NPs suspension with an equal volume of 0.03% CS solution after dialysis washing. The obtained NPs were freeze-dried (FDU-1200, Tokyo Rika Kikai Co., Ltd., Tokyo, Japan) with 250 mg sucrose and 150 mg l-glutamic acid as excipients, collected as NPs powder, and stored frozen at −30 °C until use. When NPs were used, they were redispersed in purified water.

### 3.4. Preparation of FITC-HEL-Loaded PLGA and PLGA-PEG-PLGA Nanoparticle Formulations

To measure the HEL skin retention in ex vivo studies and observe the HEL skin permeation pathway, FITC-HEL NPs were prepared by labeling HEL with the fluorescent reagent FITC. First, FITC-HEL was prepared [33,34]. Then, HEL (40 mg) was dissolved in 10 mL carbonate buffer solution (pH 9), and 1 mg FITC was dissolved in 1 mL DMSO. The labeling reaction was performed by stirring the 0.4% (*w*/*v*) HEL solution at 250 rpm and adding 0.1% (*w*/*v*) FITC solution dropwise at 25 °C. At 2 h after the drop, the reaction solution was sealed in a cellulose tube (UC36-32-100, fractional molecular weight of 14,000, pore size of 50 Å, EIDIA Co., Ltd., Tokyo, Japan). Then, 1 L of purified water was used as the outer solution for dialysis washing, which was performed as described in Section 3.3. After collecting the solution, the powder was freeze-dried, and the collected powder was used as FITC-HEL. Before use, the NPs were redispersed in purified water.

### 3.5. Evaluation of Physical Properties of PLGA and PLGA-PEG-PLGA Nanoparticle Formulations Loaded with HEL and FITC-HEL

The freeze-dried nanoparticle formulations were redispersed in purified water, and the physical properties of each nanoparticle were evaluated. The volume size distribution and average particle size of the prepared NPs were evaluated by dynamic light scattering (DLS) using a Zetasizer (ELSX-2000ZS, Otsuka Electronics Co., Ltd., Hirakata, Japan) at a skin surface temperature of 32 °C. Zeta potential measurements were performed using the same Zetasizer to evaluate the electrophoretic mobility of NPs at 32 °C in the same 5 mM NaCl solution as that used during drug administration, followed by calculation of zeta potential. The HEL content in the NPs was determined by high-performance liquid chromatography (HPLC, SIL-20AC prominence, SPD-20A prominence, LC-20AD prominence, CTO-20AC, CBM 20AC prominence, Shimadzu Corp., Kyoto, Japan) under the same conditions as described in a previous study, with an absorbance of 280 nm, a measurement time of 15 min, and a sample injection volume of 50 μL [35]. Furthermore, the HEL content and entrapment efficiency were evaluated based on the measured HEL amount and mass of NPs after freeze-drying. Then each value was calculated as follows:(1)$$Yield\;(\%)=\frac{Total\;mass\;of\;the\;prepared\;nanoparticles}{Total\;mass\;of\;the\;materials\;used\;for\;the\;nanoparticle\;formulation}\times 100$$
(2)$$HEL\;content\;in\;NPs\;(\%)=\frac{Total\;mass\;of\;the\;HEL\;contained\;in\;the\;nanoparticles}{Total\;mass\;of\;the\;prepared\;nanoparticles}\times 100$$
(3)$$HEL\;entrapment\;efficiency\;(\%)=\frac{Total\;mass\;of\;the\;HEL\;contained\;in\;the\;nanoparticles}{Total\;mass\;of\;the\;HEL\;used\;for\;the\;nanoparticle\;formulation}\times 100$$

Moreover, to confirm the release characteristics of HEL from each NP, in vitro drug release studies were conducted with Franz cells using cellulose ester tubing (Spectra Por^®^ Biotech, fractionated molecular weight 300,000, Spectrum Laboratories, Rancho Dominguez, CA, USA) as the dialysis membrane. Each NP formulation (3.0 mL) containing 0.01% (*w*/*v*) HEL adjusted to 5 mM NaCl was tested as the donor solution for Franz cells, and 19.0 mL PBS was used as the receptor solution at 32 °C. The tests were conducted with IP, and Ag/AgCl was used as the electrode, with an anode of Ag (AG-403321, The Nilaco Corporation, Tokyo, Japan) on the donor side and a cathode of AgCl (Sintered plate 0.8 t × 60 × 60, Unique Medical Co., Ltd., Tokyo, Japan) on the receptor side in the HEL-loaded CS-PLGA NPs group. In the HEL-loaded PLGA-PEG-PLGA NPs group, the cathode (AgCl) was attached to the donor side, and the anode (Ag) was attached to the receptor side. In the tests, a constant voltage of 3.0 V was applied for 2 h. Samples measuring 1.0 mL were collected from the receptor solution (0.5, 1, 2, 4, 8, 12, and 24 h), and after collection, an equal volume of PBS was added to the receptor solution. The collected samples were analyzed for HEL release by HPLC, and the cumulative HEL release was calculated.

### 3.6. Drug Skin Retention Study of FITC-HEL-Loaded PLGA and PLGA-PEG-PLGA Nanoparticle Formulations

Dorsal mice skin explants from BALB/c mice were used for the tests. Specifically, the mice were first temporarily anesthetized in a container filled with approximately 2 mL of isoflurane, followed by intraperitoneal administration of a triad of anesthetics (1.0 mL/100 g) and general anesthesia. The mice were then placed in the supine position with their limbs immobilized, hair was removed using clippers from the chest to the base of the hind legs, and the abdominal skin was excised in a circular shape with a radius of 1 cm to fit the size of the Franz cell. The excised skin was attached to a Franz cell, 3 mL of each NP suspension containing 0.025% (*w*/*v*) FITC-HEL adjusted to 5 mM NaCl was added to the donor side, and constant voltage loading was performed for 2 h under the same dosing conditions as described in Section 3.5. Two groups were tested, viz., the FITC-HEL-loaded CS-PLGA NPs group and the FITC-HEL-loaded PLGA-PEG-PLGA NPs group. At the end of the test period, the skin was collected from the Franz cells, and the HEL in the skin was extracted and quantified by HPLC. The specific skin preparation method is as follows. First, the skin was wiped with purified water after testing to remove residual FITC-HEL and NPs from the skin surface. Then, the skin was freeze-dried, cut into rice-sized pieces with scissors, and added to a test tube, after which 2 mL of extraction solvent (DMSO:purified water = 2:1) and fluorescein solution as an internal standard were added, and the tube was shaken for 60 min. The supernatant was then centrifuged at 4000 rpm for 10 min, collected, and filtered through an HPLC pretreatment filter (pore size: 0.45 µm, diameter: 13 mm, Nippon Genetics Co., Ltd., Tokyo, Japan). HEL was quantified by HPLC (SIL-20A prominence, SPD-20A prominence, LC-20AD prominence, CTO-10ASvp, DGU-20A3 prominence, Shimadzu Corp., Kyoto, Japan). The HPLC measurement conditions were set at an excitation wavelength of 495 nm and a measurement wavelength of 520 nm; the remaining measurements were performed under the same conditions as described in Section 3.5.

### 3.7. Observation of Intradermal Permeation Pathway of FITC-HEL

The skin used for ex vivo drug retention was tested using the method described in Section 3.6. It was used to observe the skin penetration pathway of FITC-HEL after administration. After the completion of tests, the skin surface was washed, stapled to a circular template, fixed, and immersed overnight at 4 °C in a fixative solution (4% (*w*/*v*) paraformaldehyde, 10% (*w*/*v*) sucrose/PBS) for tissue fixation. Then, 20-µm-thick sections were obtained using a cryostat (CM3050S, Leica Biosystems, Nussloch, Germany) attached to a cryofilm (type ⅢC (16UF), SECTION-LAB Co., Ltd., Yokohama, Japan), and the remaining embedding agent on the sections was removed using purified water and attached to glass slides [36]. After drying, the FITC-HEL remaining in the skin was observed under a confocal laser microscope (TCS SP8, Leica microsystems, Wetzlar, Germany).

### 3.8. In Vivo Percutaneous Immunization Experiments

Four groups were tested, viz., a group of healthy mice (naïve), a positive control HEL aqueous solution subcutaneous injection group (solution-SC), a group of HEL-loaded CS-PLGA NPs (CS-PLGA NPs-IP) administered percutaneously with IP, and a group of HEL-loaded PLGA-PEG-PLGA NPs (PLGA-PEG-PLGA NPs-IP) administered percutaneously with IP. Drug administration was performed on the site of hair removal in the back of mice as described in Section 3.5. Each test solution was prepared as a 0.025% (*w*/*v*) HEL solution adjusted to 5 mM NaCl. For the IP combination percutaneous group, a medical gauze of cotton material was folded in quadruplicate, cut into 1 cm × 1 cm fractions, and placed at the abdominal administration site of the mice. Then, as electrodes for IP, an Ag electrode was placed on the anode, and an AgCl electrode was placed on the cathode of the drug administration side for the CS-PLGA NPs group. For the PLGA-PEG-PLGA NPs group, an AgCl electrode was placed on the cathode, and an Ag electrode was placed on the anode of the drug administration side. Next, the electrodes were fixed with medical tape and dosed for 2 h at a constant voltage of 3 V. Drug administration was performed every two weeks four times. At two weeks after the last drug administration (eight weeks after the start of treatment), 200 µL blood was collected from the jugular vein of mice, refrigerated at 4 °C overnight, and centrifuged at 6000 rpm for 10 min at 4 °C to obtain 100 µL serum. The obtained serum was stored frozen at −80 °C until use. Treatment evaluation was performed by measuring the levels of HEL-specific IgG antibodies (IgG1 and IgG2a) and total IgE antibodies in serum by ELISA [37]. Regarding HEL-specific IgG antibodies, IgG1 and IgG2a titers were measured to confirm the activation of somatotropic and intracellular immunity, as described in previous studies [24,35]. Absorbance (O.D.) at 405 nm was measured using a plate reader (ARVO X4, PerkinElmer, Waltham, MA, USA). Serum antibody titer (ELISA antibody titer) was expressed as the maximum dilution of serum at which the O.D. was >0.1 for convenience. The total IgE antibody assay was performed according to the kit instructions (LBIS^®^ Mouse IgE ELISA Kit, Fujifilm Wako Shibayagi Corp., Gunma, Japan).

### 3.9. Data Analysis

All data are presented as the mean ± standard deviation of mean. The data were analyzed using one-way ANOVA to compare among three or more groups, followed by Tukey–Kramer tests. The Student’s *t*-test was used for comparisons between two groups. *p* < 0.05 was considered statistically significant. Analyses were performed with GraphPad Prism 7 (GraphPad Software, San Diego, CA, USA).

## 4. Discussion

Through this study, we discovered that the newly focused PLGA-PEG-PLGA nanoparticles exhibit higher therapeutic efficacy in treating food allergies in mice compared to the CS-PLGA nanoparticles studied previously. No differences were observed in the average particle size and volume size distribution of each prepared NP. Furthermore, previous studies have confirmed that PLGA and PLGA-PEG-PLGA nanoparticles, prepared using methods analogous to those used in this study, are spherical in shape [25]. The zeta potential of HEL-loaded CS-PLGA NPs was observed positively charged, due to the protonation of the amino groups in chitosan at physiological pH [38]. In contrast, the charge of HEL-loaded PLGA-PEG-PLGA NPs is negative due to the carboxyl group at the PLGA end [39,40]. The presence of uncharged PEG chains may have resulted in a relatively low negative absolute zeta potential of the HEL-loaded PLGA-PEG-PLGA NPs [39]. The lower HEL content in the HEL-loaded PLGA-PEG-PLGA NPs than in the HEL-loaded CS-PLGA NPs may be attributed to the tendency of hydrophilic drugs to diffuse into the aqueous phase during intermixing during particle preparation, particularly in PLGA-PEG-PLGA NPs, where the drug is not fully retained in the NPs due to the hydrophilic PEG exposed on the particle surface [41,42]. The results of the release studies also suggest that the PEG present on the surface of the HEL-loaded PLGA-PEG-PLGA NPs facilitated the leakage of the encapsulated drug and a faster release. Similarly, in a previous study, it was reported that PLGA-PEG-PLGA NPs exhibited a faster drug release rate than PLGA NPs [25]. The results of ex vivo measurement of intracutaneous HEL accumulation and observation of the intradermal permeation pathway of FITC-HEL demonstrated that the FITC-HEL-loaded PLGA-PEG-PLGA NPs exhibited a higher HEL skin accumulation and HEL delivery to the deep hair follicle compared to the FITC-HEL-loaded CS-PLGA NPs. Generally, following the transdermal administration of NPs, proteins present in the skin may adhere to the particles, resulting in aggregation and a reduction in the stability of NPs. This may inhibit the transdermal penetration of drugs [11]. However, in HEL-loaded PLGA-PEG-PLGA nanoparticles, the surface of the nanoparticles formed a hydrophilic layer derived from PEG chains, which may have prevented the adhesion of proteins and maintained stability [43]. In in vivo treatment experiments, titers of anti-HEL antibodies, IgG1, IgG2a, and IgE, were measured eight weeks after the start of treatment. In food allergies, IgE antibodies induce immediate anaphylaxis by interacting with allergens. On the other hand, IgG binds to the inhibitory FcγRIIb receptor, and is known to prevent IgE-mediated anaphylactic food allergy and restore desensitization [44,45]. IgG1 antibody production was significantly increased in the PLGA-PEG-PLGA NPs group compared to the CS-PLGA NPs group. Antigen-presenting cells are known to be distributed in large numbers around hair follicles and to uptake antigens through the follicle [46,47]. As shown in results 3.2 and 3.3, the HEL-loaded PLGA-PEG-PLGA NPs may have exhibited superior efficiency in allergen exposure compared to HEL-loaded CS-PLGA NPs by facilitating the delivery of HEL to the deep hair follicles. Furthermore, the PLGA-PEG-PLGA NPs group may have increased the production of IgG1 antibodies and decreased the production of IgE antibodies due to the adjuvant effect of PEG in the NPs accumulated in the skin [48]. Therefore, transdermal administration of HEL-loaded PLGA-PEG-PLGA NPs in combination with iontophoresis is a more effective method of administration, as it induces a decrease in IgE antibody production and an increase in IgG antibody production.

The PLGA-PEG-PLGA nanoparticles in this study demonstrated faster drug release and higher intradermal drug retention compared to PLGA nanoparticles. Consistent with previous reports, this reaffirms the utility of PLGA-PEG-PLGA nanoparticles in transdermal drug delivery [25]. This study was conducted using healthy mouse models. As a future direction, it is necessary to confirm whether PLGA-PEG-PLGA nanoparticles have the same effect on mice models of allergic pathology as in the present study. Furthermore, it is important to note that the therapeutic results obtained in mice in this study may differ when applied to humans, considering factors such as the number of hair follicles. This consideration is crucial for the future clinical application of our transdermal delivery system. To verify the transdermal absorption effect on human skin, it is also a future research task to apply this transdermal delivery system to three-dimensional human skin models and evaluate its effectiveness.

## 5. Conclusions

HEL-loaded CS-PLGA NPs and HEL-loaded PLGA-PEG-PLGA NPs were prepared. Ex vivo intracutaneous HEL accumulation tests and FITC-HEL permeation pathway observation images revealed that FITC-HEL-loaded PLGA-PEG-PLGA NPs exert a higher drug percutaneous absorption enhancement effect on the percutaneous absorption pathway through the hair follicle than FITC-HEL-loaded CS-PLGA NPs. This may be attributed to the suppression of protein adhesion in PLGA-PEG-PLGA NPs and the higher stability of the NPs. The in vivo percutaneous immunization test demonstrated that total IgE antibody titers were significantly reduced in each NP group compared with those in the positive control aqueous solution subcutaneously administered group. Moreover, HEL-specific IgG1 antibody titers were significantly higher in the PLGA-PEG-PLGA NPs group than in the CS-PLGA NPs group. This may be due to the HEL-loaded PLGA-PEG-PLGA NPs penetrating deeper into the hair follicles, resulting in more efficient delivery to antigen-presenting cells distributed around the follicles. These data suggest that this percutaneous immunization system using PLGA-PEG-PLGA as a polymer in combination with nanoparticulation of HEL and IP is more effective. The results obtained in this study were tested on healthy mice. The results of this study might contribute to the development of a percutaneous immunotherapy that is more effective, safer, and less burdensome by applying it to allergy treatment experiments using pathological models and to new antigen model proteins.

## Figures and Tables

**Figure 1 molecules-29-04123-f001:**
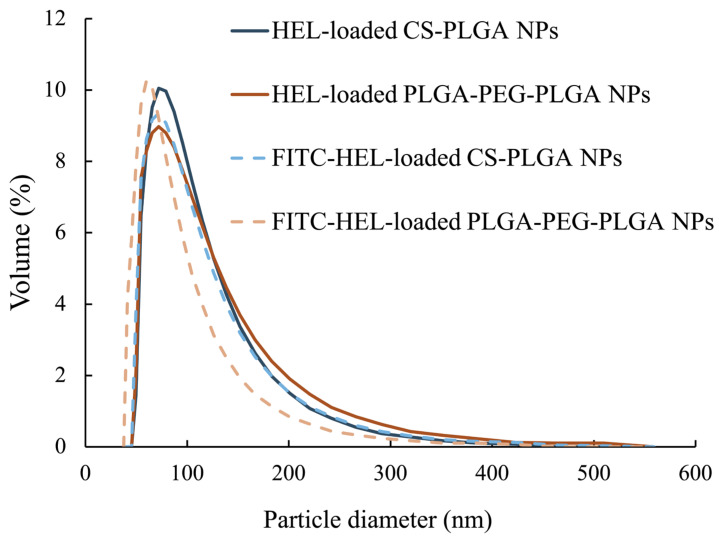
Particle size distribution of HEL-loaded CS-PLGA NPs and HEL-loaded PLGA-PEG-PLGA NPs (*n* = 3).

**Figure 2 molecules-29-04123-f002:**
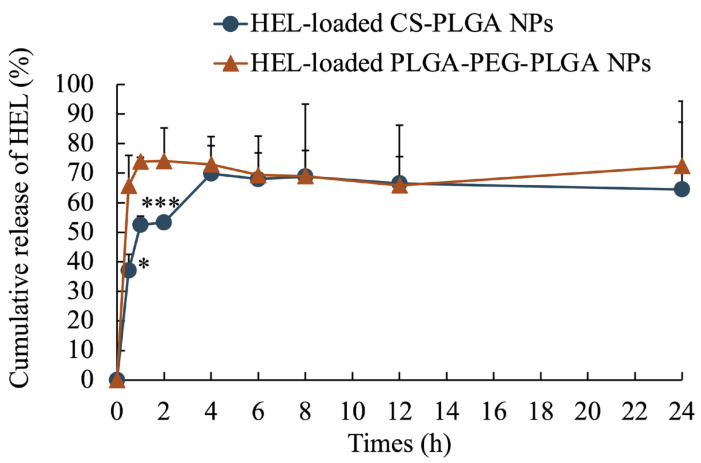
Cumulative release ratio of HEL from HEL-loaded CS-PLGA NPs and HEL-loaded PLGA-PEG-PLGA NPs. *n* = 3; data are expressed as mean ± standard deviation. *t*-test, * *p* < 0.05, *** *p* < 0.001.

**Figure 3 molecules-29-04123-f003:**
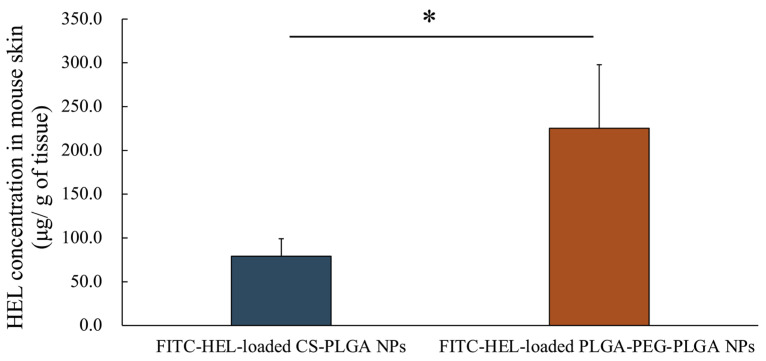
HEL concentration in rat skin after the application of FITC-HEL-loaded PLGA NPs and FITC-HEL-loaded PLGA-PEG-PLGA NPs at 2 h from the initiation of the ex vivo skin permeability tests. *n* = 3–4; data are expressed as mean ± standard deviation. *t*-test, * *p* < 0.05.

**Figure 4 molecules-29-04123-f004:**
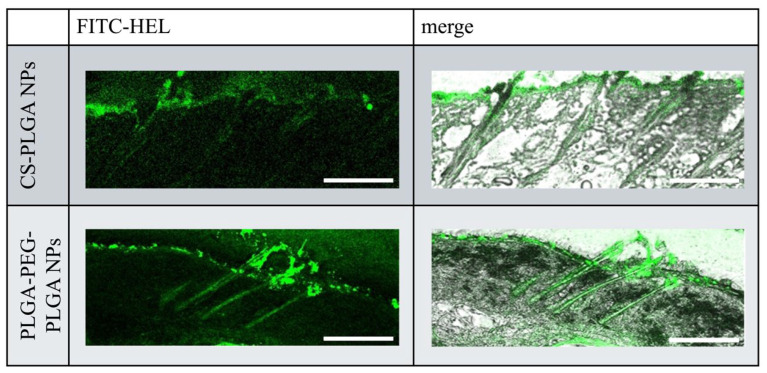
Fluorescence microscopy images of cross sections of mouse skin. Scale bar, 20 µm.

**Figure 5 molecules-29-04123-f005:**
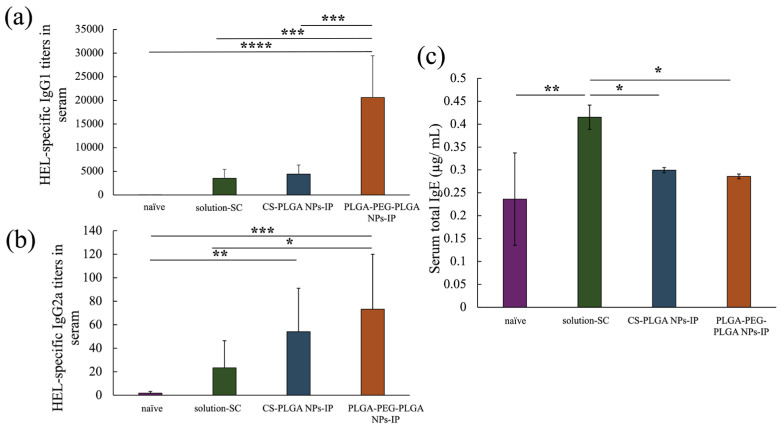
HEL-specific IgG1 (**a**), IgG2a (**b**) antibody titers in serum and total IgE concentration in serum (**c**) of naïve group, solution-SC group, CS-PLGA NPs-IP group, and PLGA-PEG-PLGA NPs-IP group. *n* = 3–7; data are expressed as mean ± standard deviation. Tukey–Kramer test, * *p* < 0.05, ** *p* < 0.01, *** *p* < 0.001, **** *p* < 0.0001.

**Table 1 molecules-29-04123-t001:** Physical properties of HEL-loaded CS-PLGA NPs and HEL-loaded PLGA-PEG-PLGA NPs. *n* = 3; data are expressed as mean ± standard deviation.

	HEL-Loaded CS-PLGA NPs	FITC-HEL-Loaded CS-PLGA NPs	HEL-Loaded PLGA-PEG-PLGA NPs	FITC-HEL-Loaded PLGA-PEG-PLGA NPs
Mean diameter (nm)	100 ± 47	99 ± 49	105 ± 54	87 ± 43
Zeta potential (mV, *I* = 5 mM)	38.4 ± 1.9		−7.2 ± 0.1	
Yield (%)	82.9 ± 12.0		67.7 ± 2.7	
HEL content in NPs (%)	8.0 ± 0.8	7.9 ± 0.7	3.4 ± 0.4	3.4 ± 0.9
HEL entrapment efficiency (*w*/*w*%)	66.7 ± 7.0	65.5 ± 6.1	28.7 ± 3.4	28.1 ± 7.8
Polydispersity index	0.23 ± 0.01	0.25 ± 0.02	0.27 ± 0.01	0.30 ± 0.01

## Data Availability

The raw data supporting the conclusions of this article will be made available by the authors on request.

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
