# Peer review of "Application of PLGA-PEG-PLGA Nanoparticles to Percutaneous Immunotherapy for Food Allergy"

_molecules, 2024, doi:10.3390/molecules29174123_

Round 1
Reviewer 1 Report
Comments and Suggestions for Authors
The manuscript is well-structured, with methods and results presented in good scientific English. Although the drug carrier (PLGA-PEG-PLGA nanoparticles) is known in the literature, the incorporation of an antigen protein in the formulated nano-structures for percutaneous immunotherapy provides novelty to the work. I appreciate the comparison that the authors make with their previous formulation, which demonstrates the search for a drug carrier with improved characteristics. My recommendations to the authors are as follows:
1. In section 2.5. – the equations used to determine HEL content and content efficiency (drug loading (DL) and entrapment efficiency (EE)) should be provided.
2. Section 2.5 – “The volume size distribution and average particle shape of the prepared NPs were evaluated by dynamic light scattering (DLS)” – average particle shape or average particle size?
3. Visualization of the formulated structures is missing (in order to determine their shape and to confirm their size). I suggest the authors to add a SEM or TEM images of the particles.
4. Are the size analysis performed before freeze drying or after freeze drying and redispersing the particles?
5. Is it possible for an interaction to occur between the polymers and the drug that would alter the drug's activity or limit its release from the matrix? А FTIR analysis could confirm the absence of such interactions.
6. Figure 2 - In my opinion, the cumulative drug release should be expressed as a percentage (%) of the amount of drug released compared to the initial amount. This way, we can evaluate not only the rate at which the drug is released, but also the extent of the drug release.
Reviewer 2 Report
Comments and Suggestions for Authors
The manuscript submitted by Akiyoshi Saitoh et al. describes studies directed toward the development of the basis of new drug formulation for allergen immunotherapy related to food allergy. The problem is important, since various kinds of allergies, among which food allergy occur very often. The approach proposed by the authors consists of the percutaneous administration of the allergen-loaded PLGA-PEO-PLGA nanoparticles, the treatment that may result in the alleviation or complete elimination of the allergy symptoms. Since the percutaneous administration is well tolerated by patients and PLGA-PEO-PLGA nanocarriers provide better control of the immune response, the design of the delivery systems made by the authors should be appreciated. Also selection of the hen egg lysozyme (HEL) as a model allergen does not cause any objection. However, before I would be ready to recommend the publication of this work the authors should answer a few questions related to conducting the research. It is rather obvious that in the high-quality research, all substrates and products should be carefully characterized and the results of the study should be presented to the readers.
1. For the preparation of nanocarriers, authors used commercial PLGA and PLGA-PEO-PLGA copolymers. The producers provided some information on these compounds, however, this information is insufficient. Authors should ask the producer how molar composition and molar mass were measured and whether the given molar masses were number or weight averaged. The question about the method used for the molar mass determination is not trivial because only some of these methods give the correct values. For example, the GPC method gives he real data only when the Multi-Angle- Light-Scattering detectors are used. The GPC with RI detectors and calibration on the polystyrene standards gives for PLA Mn about two times higher than the real one.
2. Author, please provide information on method selected for analysis of the autocorrelation function in the DLS studies of the nanocarrier nanoparticles
3. In table 1 there are given the average values of selected physical parameters of the nanoparticles presented as mean ± standard deviation. Please, justify the number of the meaningful digits used in the presentation. In my opinion they are too many. The accuracy of the means and standard deviations are given should be no larger than the standard deviation itself. Thus, for example, I suggest replacing 99.9 ± 46.7 with 100 ± 50.
Reviewer 3 Report
Comments and Suggestions for Authors
The authors have done much work synthesising and applying PLGA-PEG-PLGA nanoparticles for transdermal drug delivery. The current manuscript focuses on percutaneous immunotherapy for food allergy. However, some drawbacks as below should be addressed to improve its quality before being accepted to be published.
1. In the Introduction, since the application focuses on food allergy, the authors should introduce more research on the background of the treatment strategies of food allergy to highlight the advantages of PLGA-PEG-PLGA. What advantages do PLGA-PEG-PLGA nanoparticles have? what problems they can solve? particularly considering you have done lots of work on transdermal drug delivery by using PLGA-PEG-PLGA.
2. The authors should present the morphology characterization of the synthesized nanoparticles, like SEM and optical photographs.
3. Why did the authors synthesize PLGA nanoparticles, aside from PLGA-PEG-PLGA? The purposes and research significance should be highlighted in the Introduction.
4. Likewise, why did you prepare FITC-HEL-loaded nanoparticles except for HEL-loaded nanoparticles? They may be meaningless but just add your workload as long as you can indicate the research gap and significance in the Introduction.
5. In the Discussion, line 368-370, the authors claimed that "PEG coating on the surface of the ... nanoparticles facilitated the leakage of ...: What is the mechanism of PEG coating in the leakage of nanoparticles? How did you coat PEG on the surface of nanoparticles?
Round 2
Reviewer 1 Report
Comments and Suggestions for Authors
The authors have made the suggested corrections and have answered my questions. I have no further recommendations.